# Enhanced Lung Cancer Survival Prediction Using Semi-Supervised Pseudo-Labeling and Learning from Diverse PET/CT Datasets

**DOI:** 10.3390/cancers17020285

**Published:** 2025-01-17

**Authors:** Mohammad R. Salmanpour, Arman Gorji, Amin Mousavi, Ali Fathi Jouzdani, Nima Sanati, Mehdi Maghsudi, Bonnie Leung, Cheryl Ho, Ren Yuan, Arman Rahmim

**Affiliations:** 1BC Cancer Research Institute, Vancouver, BC V5Z 1L3, Canada; cho@bccancer.bc.ca (C.H.); arman.rahmim@ubc.ca (A.R.); 2Department of Radiology, University of British Columbia, Vancouver, BC V6T 1Z4, Canada; ren.yuan@bccancer.bc.ca; 3Technological Virtual Collaboration (TECVICO Corp.), Vancouver, BC V6L 1L7, Canada; gorji.arman@edu.umsha.ac.ir (A.G.); amin.mousavi.academic@gmail.com (A.M.); a.fathi@edu.umsha.ac.ir (A.F.J.); sanatinima20@gmail.com (N.S.); dr.mmaghsudi@gmail.com (M.M.); 4Neuroscience and Artificial Intelligence Research Group (NAIRG), Department of Neuroscience, Hamadan University of Medical Sciences, Hamadan 6517838736, Iran; 5BC Cancer, Vancouver Center, Vancouver, BC V5Z 1L3, Canada; bonnie.leung@bccancer.bc.ca; 6Department of Physics & Astronomy, University of British Columbia, Vancouver, BC V6T 1Z4, Canada

**Keywords:** lung cancer, deep and handcrafted radiomic features, machine learning, survival prediction, supervised and semi-supervised strategy

## Abstract

This study presents a novel semi-supervised learning (SSL) approach that improves lung cancer survival predictions by incorporating diverse datasets, including head and neck cancer (HNCa), alongside handcrafted and deep radiomic features (HRF/DRF) from PET/CT scans. By shifting from traditional supervised learning to SSL, our method addresses data limitations by using heterogeneous yet clinically relevant data, achieving an average accuracy of 0.85 ± 0.05 with PCA + Multi-Layer Perceptron (MLP) models. This approach not only enhances predictive performance but also introduces a new paradigm for leveraging diverse datasets in tasks with limited data.

## 1. Introduction

Lung cancer (LCa) is a global health issue, with 2.21 million new cases and over 1.8 million annual deaths projected by 2030, making it the second-most-diagnosed and leading cause of cancer deaths [1,2,3,4]. Key challenges include the lack of effective biomarkers and delayed diagnosis, along with insufficient funding [5,6]. Accurate prognostic models are needed to enhance clinical decision-making, with overall survival (OS) often used to assess treatment efficacy [7,8,9]. While supervised learning (SL) methods are effective, acquiring labeled data is costly and difficult [10,11]. Semi-supervised learning (SSL), which uses limited labeled data with large unlabeled datasets, is valuable when labeled data are scarce [12,13,14,15]. SSL has shown promise in LCa prediction, outperforming fully supervised models [16], and it has enhanced diagnostics in malignant diseases [17]. Recent studies using SSL on chest X-rays and CT scans have achieved state-of-the-art lung nodule detection [16].

Head and neck cancer (HNCa) and LCa share clinical and biological similarities, including common risk factors and genetic mutations, making HNCa data useful for improving LCa survival predictions [18]. Both cancers are linked to tobacco and alcohol use and exhibit comparable tumor microenvironments and treatment responses [19]. Survivors of HNCa face a higher risk of developing second primary cancers, including LCa, with 70–80% of HNCas associated with prior tobacco use [20]. Additionally, both HNCa and LCa datasets include squamous cell carcinoma (SCC), while LCa datasets also feature data from other non-small-cell lung carcinoma (NSCLC) such as adenocarcinoma (ADC), all of which share similar radiomic features [21,22]. Incorporating HNCa data into an SSL framework allows us to enhance feature space representation, improving LCa outcome predictions. This connection between the two diseases supports using their images to improve OS predictions in LCa patients.

Positron emission tomography (PET) and computerized tomography (CT) are crucial diagnostic tools for detecting cancerous lesions and metabolic diseases. PET provides functional information about tissue metabolism, while CT offers detailed structural information [23,24]. To employ machine learning (ML) algorithms in predictive performance, one may extract imaging features such as deep radiomic features (DRF) and handcrafted radiomic features (HRF) [25]. These quantitative features are valuable for disease diagnosis, therapy response, and prognosis prediction [26,27]. HRFs are manually generated features, capturing known aspects like shape, intensity, and wavelet characteristics [28]; deep radiomic features (DRFs), on the other hand, are quantitative imaging biomarkers automatically extracted using deep learning models, such as convolutional neural networks (CNNs), capturing high-dimensional and hierarchical patterns in medical images for tasks like diagnosis and prognosis, but they often lack clear interpretability and clinical validation [29,30]. Although DRFs provide deeper insights, HRFs offer greater interpretability and reproducibility [31,32,33,34,35]. Combining DRFs and HRFs provides complementary information, enhancing image analysis [36].

A hybrid machine learning system (HMLS) combines different ML techniques to maximize their strengths and minimize limitations [37,38,39,40]. HMLSs are widely used in medical diagnostics to improve prediction accuracy and reliability [41,42,43]. For example, integrating DL with Bayesian networks enhances prognosis forecasting for LCa patients [44]. DL extracts complex features from large datasets, while Bayesian networks incorporate prior knowledge and uncertainty, resulting in more accurate and interpretable outcomes [45]. HMLSs effectively address challenges such as missing data, noise, and diverse data sources in medical analysis [46]. Principal component analysis (PCA) reduces dimensionality in large datasets, improving interpretability and minimizing information loss by creating new uncorrelated variables (principal components) that maximize variance [47,48]. This enhances model performance and generalizability, facilitating pattern identification and prediction accuracy, which this study aims to achieve.

Survival analysis is a statistical method used to predict the time until an event, such as disease onset or death, occurs [49]. It addresses time-to-event data and censored data, compares survival curves, and explores relationships between variables and survival time. Techniques like Cox’s proportional hazards regression allow for multiple covariates and generate hazard ratios for factors [50,51,52]. While some studies have examined survival analysis for outcome prediction [53,54,55], we propose a method that categorizes OS into two classes: class 1 for patients surviving over two years (the average survival time in LCa datasets) and class 2 for those deceased within two years. This approach, combined with survival regression algorithms (SRAs), improves prediction accuracy and helps understand the relationship between features and outcomes for new patients.

This study explores the benefits of using diverse datasets, including HNCa, in an SSL approach with pseudo-labeling, alongside LCa datasets, to improve prediction performance compared to SL focused only on LCa. Key focuses include the following: (i) addressing the challenge of creating large labeled datasets by utilizing SSL with limited labeled LCa data and a large amount of unlabeled HNCa data to predict OS in LCa, (ii) comparing CT images with PET for their wider availability and lower cost, (iii) comparing DRFs and HRFs for improved prediction accuracy, (iv) evaluating SL using four binary-classification algorithms (CAs) with PCA on HRF and DRF datasets from 199 LCa PET/CT images, expanding with 408 pseudo-labeled HNCa cases, and (v) investigating four hazard ratio survival analysis algorithms (SRAs) with PCA for a deeper understanding of survival time and time-to-event patterns.

## 2. Materials and Methods

### 2.1. Patient Data and Image Preprocessing

We utilized a dataset of 199 LCa patients obtained from The Cancer Imaging Archive (TCIA) and the BC Cancer Agency, a publicly funded provincial comprehensive cancer care program serving approximately 5.2 million residents in British Columbia, Canada. This dataset includes various LCa subtypes, such as SCC, ADC, NSCLC not otherwise specified (NOS), and other subtypes, with stages ranging from IA to IIB. For head and neck cancer (HNCa), we included 408 patients with SCC at stages ranging from I to IVC, along with PET and CT imaging data sourced from TCIA. We only selected patients who had both CT and PET. We then selected patients whose PET images had AC (Attenuation Correction). AC is a process used to compensate for the loss of photons as they travel through the body [56]. These selection criteria yielded an LCa group of 33 from TCIA subjects (14 males, 19 females), 166 BC cancer subjects (85 males, 81 females), and 408 HNCa from TCIA (320 males and 88 females). Table 1 presents the characteristics of the patients across the different datasets. A more detailed description of the demographics specific to the LCa patients can be found in Appendix A. Binary OS was considered as an outcome, categorizing them into two classes: class 1 (survivors), alive after two years (averaged death time in LCa dataset), and class 2 (non-survivors), deceased within two years of diagnosis. In the pre-processing step, PET images were first registered to CT. Subsequently, Standardized Uptake Value (SUV) correction was performed for the raw PET data, which enables standardized measurements of tracer uptake in medical imaging studies [57]. Moreover, the clipping technique scales lung cancer CT images by restricting intensity values to a range, typically from −300 to 300 Hounsfield units (HU). Finally, the minimum/maximum (min/max) normalization technique was applied to both images to scale them [58]. Figure 1 shows that unlike some studies [59,60] that have used automatic segmentation, this work relied on collaborative physicians to manually delineate cancer-affected areas on the PET/CT images for imaging feature extraction.

### 2.2. Study Procedure

The present study aimed to predict binary OS in patients with LCa, as shown in Figure 2. After image preprocessing and mask delineations (parts i and ii in Figure 2), as outlined in Section 2.1 and illustrated in Appendix A, two imaging feature extraction methods—HRF and DRF—were employed for quantitative analysis (part iii). A total of 215 standardized HRFs were obtained using the standardized PySERA module in the ViSERA 1.0.0 software (visera.ca, 1 January 2024) [61]. As shown in Appendix A, these features included 20 first-order features, 30 intensity histogram features, and 136 texture features containing co-occurrence matrixes (50 features), run-length matrices (32 features), size zones (16 features), distance zones (16 features), neighborhood gray-tone difference matrices (5 features), and neighboring gray-level dependence matrixes (17 features). Additional details on these HRFs can be found in Appendix A.

Similarly, 1024 DRFs were derived from the segmented PET and CT images from the bottleneck layer of the 3D autoencoder embedded within the ViSERA software. As shown in Appendix A, A typical 3D autoencoder comprises an encoder and a decoder network. The encoder compresses input images into a latent representation, while the decoder reconstructs the original images. The input and output layers have equal neurons, with identical input data and labels. The network includes three 3 × 3 convolutional layers, 2 × 2 max-pooling, and LeakyReLU activations. Pooling layers reduce parameters, while the decoder employs three 3 × 3 convolutional layers, LeakyReLU, and up-sampling. The autoencoder was trained using binary cross-entropy loss with the Adam optimizer. Finally, 1024 DRFs were extracted from the bottleneck layer using segmented CT and PET images. The architecture of the autoencoder is fully explained in Appendix A. After DRF and HRF generation, 199 LCa datasets were split into 80% for five-fold cross-validation and 20% for external nested testing (part iv). Additionally, stratification was performed to ensure that both divisions contained all outcomes. In addition, all extracted features were normalized by a min–max function (part v). Moreover, different combinations of imaging feature sets such as DRF-CT, DRF-PET, HRF-CT, HRF-PET, DRF-CT plus HRF-CT, and DRF-CT plus HRF-CT were employed in this study.

For prediction, we employed two strategies—SL and SSL—within an HMLS framework, which integrated PCA with classification algorithms (CAs) or SRAs (part vi). By combining multiple approaches, the HMLS leverages the strengths of each method while mitigating their weaknesses [62]. For the CAs, we used six classifiers: Multi-Layer Perceptron (MLP) [63], Support Vector Machine (SVM) [64], K-Nearest Neighbor (KNN) [65], Ensemble Voting (EV), Extreme Gradient Boosting (XGB), and Light Gradient-Boosting Machine (LGB) (part vii). For the SRAs, we utilized Fast Survival SVM (FSVM) [66], Component-wise Gradient Boosting Survival Analysis (CWGB) [67], Random Survival Forest (RSF) [68], and Cox’s regression (COXR) (part viii) [69]. Hyperparameters for all classifiers were optimized using a grid search, as detailed in Appendix A, which significantly enhances the performance of ML algorithms. More detailed explanations of PCA, CAs, and RSAs are provided in Appendix A. In PCA, we aimed to retain about 90% of the variance. Based on this criterion, as shown in Appendix A, we determined that selecting 10 components was sufficient to capture this level of variance across the different datasets. In our study, we used a DL network (3D autoencoder) solely to extract quantitative DRFs from imaging data. These DRFs were then used to train ML classifiers for survival prediction. By limiting the role of DL to feature extraction, we ensured that prediction tasks were handled entirely by ML algorithms, maintaining a clear distinction between feature extraction and prediction processes.

The SSL strategy incorporated a pseudo-labeling procedure, which utilized both labeled and unlabeled data. Pseudo-labeling, as explained in Appendix A, involves training a model on labeled data to generate predictions for unlabeled data. These predictions, referred to as “pseudo-labels”, are then used to train the model on the previously unlabeled data in an SL manner. In this approach, a Random Forest (RandF) algorithm was employed to label the HNCa cases based on the training LCa data, and these pseudo-labeled HNCa cases were added to the training LCa dataset. During five-fold cross-validation, 80% of the 199 LCa datasets were divided into five folds. Four folds were used for training, while one fold was used for validation. The RandF algorithm was trained on four training folds and used to label 408 HNCa datasets. The combined dataset of labeled HNCa samples and the four training folds was then input into six HMLSs (PCA + CAs) models and tested on the remaining fold. Importantly, the validation and external testing sets were never used in the training or labeling process. SSL expanded the dataset by incorporating 408 pseudo-labeled HNCa cases alongside 199 LCa cases. In contrast, the SL strategy used only the 199 labeled LCa PET and CT images, applying the same HMLS methods. We then compared the effectiveness of the expanded SSL with the SL methods.

For survival hazard ratio analysis and prediction, we employed four HMLSs (PCA + SRAs). The survival analysis used the median technique to categorize samples into low- and high-risk groups, training the SRAs with these categories alongside their continuous time data. A total of 199 LCa patients were included. Since the SSL strategy is unsuitable for survival analysis due to the requirement of the last follow-up time, which is unpredictable, we exclusively used an SL strategy, relying on data from the 199 LCa patients. Therefore, OS data or last follow-up times were not available for the HNCa patients in this study. Moreover, we utilized High-Dimensional Hotelling’s T Squared (HDTS) test for a further analysis of the two groups, including low- and high-risk patients [70]. Furthermore, we used the mean function to determine high- and low-risk factors for patients, so the patients who had a death time over the average death time were considered as low-risk and the patients with a death time lower than the average were considered as high-risk. This method compared the meaning of two populations (high- and low-risk) when the feature number (F) of the components was larger than the sample number (N), as elaborated in Appendix A. Thus, the HDTS test was applied to all LCa datasets, including DRF-CT, DRF-PET, HRF-CT, and HRF-PET. Additionally, the average accuracy from the five-fold cross-validation and external nested testing were calculated to evaluate the models in the classification task. The C-index and log rank *p*-value were used to compare the survival risk assessments. The Kaplan–Meier survival analysis method [52] was used to examine the difference in OS between patients categorized as high-risk or low-risk by the model, and the differences between groups were assessed by the log rank test [51].

## 3. Results

This study investigated both the SL and SSL strategies. Furthermore, two kinds of imaging features, namely HRFs and DRFs, were extracted from both PET and CT images. In addition, we used segmented masks to extract DRFs and HRFs from the images. Furthermore, different HMLSs were employed to predict OS through the provided datasets using PCA combined with both CAs and SRAs.

### 3.1. Result Provided Using Classification Analysis

#### 3.1.1. Results Provided Using HRF Sets

In the HRF strategy, we first employed HRFs extracted from both PET and CT images as well as the HMLSs mentioned in Section 2. Figure 3 illustrates that the SL method, incorporating HRF-CT, PCA, and LGB, achieved the highest average accuracy, with a score of 0.71 ± 0.04 and an external nested test accuracy of 0.64 ± 0.05. Conversely, the strategy using HRF-PET, PCA, and XGB attained a slightly lower performance of 0.69 ± 0.07, with an external nested test result of 0.62 ± 0.02. Furthermore, the use of PCA, XGB algorithms, and HRF-CT, with an average accuracy of 0.70 ± 0.06 and an external nested test accuracy of 0.63 ± 0.02, did not add value to the prediction tasks in the SL strategy.

In the SSL strategy, the combination of HRF-PET, PCA, and MLP achieved the highest average accuracy of 0.77 ± 0.10 and an external nested test score of 0.72 ± 0.02. Furthermore, the area under the curve (AUC) metric was approximately 0.69 for both the five-fold cross-validation and the external test. The confusion matrix and ROC curves are shown in Appendix A. Meanwhile, using HRF-CT with PCA and XGB resulted in an average accuracy of 0.76 ± 0.06 and an external nested test score of 0.62 ± 0.01. Additionally, other combinations, such as HRF-PET with PCA plus both SVM and KNN, achieved an average accuracy exceeding 0.76, while the combination of HRF-CT with PCA plus both MLP and LGB reached an average accuracy of 0.75 in the SSL strategy. Furthermore, combining PCA and either HRF-CT or HRF-PET in the SSL strategy, using SVM and XGB, with average accuracies of 0.73 ± 0.07 and 0.74 ± 0.05, respectively, added no value to the classification tasks. Additionally, these HMLSs achieved an external nested test accuracy of 0.68 ± 0.0, and 0.67 ± 0.05, respectively. Overall, the SSL methods using the HRF frameworks significantly outperformed the other methods, drawing on data from both PET and CT images (*p* = 0.01, paired *t*-test), compared to the top performance in the SL strategy, which was 0.71 ± 0.04, achieved by a combination of HRF-CT, PCA, and LGB. All the nested external testing performances are shown in Appendix A.

#### 3.1.2. Results Provided Using DRF Sets

Within the DRF framework, DRFs extracted from both PET and CT images were employed alongside the HMLSs outlined in Section 2. Figure 4 shows that the SL strategy using DRF-CT, PCA, and LGB achieved the highest average accuracy, registering at 0.69 ± 0.06, with an external nested test accuracy of 0.57 ± 0.04. On the other hand, the strategy utilizing DRF-PET with the same HMLS setup recorded a lower average accuracy of 0.62 ± 0.09, but it resulted in a better external nested test score of 0.68 ± 0.03. Furthermore, the mixture of PCA, XGB algorithms, and DRF-CT, with an average accuracy of 0.67 ± 0.01 and an external nested test accuracy of 0.58 ± 0.06, did not contribute to improving classification tasks in the SL strategy.

In the SSL domain, the DRF-PET, PCA, and MLP configuration achieved the highest average accuracy, recording 0.85 ± 0.05, with an external nested test score of 0.80 ± 0.01. Moreover, the AUC was approximately 0.75 for both the five-fold cross-validation and the external test. The confusion matrix and ROC curves are shown in Appendix A. Similarly, the DRF-CT setup with the same HMLS configuration attained an average accuracy of 0.83 ± 0.06 and an external nested test accuracy of 0.79 ± 0.01. The additional SSL configurations, involving other HMLSs paired with DRF-PET or DRF-CT combined with PCA and both SVM and KNN, exceeded an average accuracy of 0.81. In summary, although the combination of DRF-PET, PCA, MLP, and the SSL strategy achieved the highest average accuracy of 0.85, there was no significant difference in performance between the SSL strategy combined with PCA and the three classifiers (MLP, SVM, and KNN) for both DRF-PET and DRF-CT. This indicates that these algorithms within the SSL strategy performed effectively. Collectively, the SSL strategies leveraging the DRF framework significantly outperformed the highest SL result of 0.69 ± 0.06, achieved by DRF-CT, PCA, and LGB, as demonstrated by data from both the PET and CT images and validated by a *p* << 0.001 from the paired *t*-test. Moreover, the superior performance of 0.85 ± 0.05 achieved by DRF-PET using the SSL strategy significantly surpassed the highest performance of 0.77 ± 0.10 achieved with HRF-PET when paired with PCA and MLP, as confirmed by a *p* = 0.003 in the paired *t*-test. All the nested external testing performances are shown in Appendix A.

#### 3.1.3. Results Provided Using a Mixture of DRF and HRF Sets

In the SL strategy that combined the HRF and DRF approaches, as illustrated in Figure 5, the incorporation of HRF-CT and DRF-CT with PCA and KNN yielded an average accuracy of 0.63 ± 0.09 and an external nested test accuracy of 0.57 ± 0.05. Additionally, the strategy using HRF-PET combined with DRF-PET, PCA, and LGB achieved a slightly lower average accuracy of 0.62 ± 0.06 but demonstrated a better external nested test accuracy of 0.69 ± 0.07. Furthermore, employing HRF-CT along with DRF-CT, PCA, and LGB resulted in a comparable average accuracy of 0.62 ± 0.0, with an improved external nested test score of 0.74 ± 0.06. The differences in performance between these strategies were not statistically significant.

In the SSL strategy, the combination of HRF + DRF (extracted from PET), PCA, and MLP achieved the highest average accuracy of 0.78 ± 0.09 and an external nested test score of 0.75 ± 0.01. Meanwhile, the AUC was approximately 0.72 for both the five-fold cross-validation and the external test. The confusion matrix and ROC curves are shown in Appendix A. Additionally, the other combinations, such as HRF+ DRF (extracted from PET), PCA, and either SVM, KNN, or EV, also achieved average accuracies exceeding 0.76, with no significant difference between these performances. Similarly, the SSL strategy using HRF-CT and DRF-CT with PCA and either MLP or SVM achieved an average accuracy of 0.74 ± 0.05 and an external nested test score above 0.70. Other combinations, such as the incorporation of HRF + DRF (extracted from CT) with either PCA + KNN or PCA + EV, achieved average accuracies exceeding 0.72, with no significant difference between these performances. Overall, the SSL strategy using HRF and DRF for both CT and PET significantly outperformed the SL strategy (*p* << 0.001, paired *t*-test). All nested external test performances are detailed in Appendix A.

### 3.2. Results Provided Using Survival Analysis

The HDTS test indicated significant differences between DRFs extracted from CT and PET in both the high-risk and low-risk groups, with *p*-values of 0.023 and 0.016, respectively. In contrast, there were no significant differences between HRFs extracted from CT and PET in the high-risk and low-risk groups, with *p*-values of 0.452 and 0.577, respectively. Subsequently, survival analysis was performed across the six datasets, i.e., HRF-CT, HRF-PET, DRF-CT, DRF-PET, a mixture of DRF-PET and HRF-PET, and a mixture DRF-CT and HRF-CT, using PCA combined with four SRAs. As depicted in Figure 6, the HRF frameworks utilizing PCA + CWGB on HRF-CT achieved an average C-index of 0.79 ± 0.08 and a log rank *p*-value significantly less than 0.001, significantly outperforming the same algorithms applied on HRF-PET, which had an average C-index of 0.62 ± 0.05 and a log rank *p*-value of 0.05 (*p* << 0.001, paired *t*-test). Additionally, these models recorded nested external testing C-indexes of 0.80 ± 0 and 0.59 ± 0.03, respectively. In the DRF model, PCA + CWGB on DRF-CT resulted in an average C-index of 0.80 ± 0.10 and a log rank *p*-value significantly less than 0.001, significantly outperforming the same algorithms applied on DRF-PET, which had an average C-index of 0.53 ± 0.09 and a log rank *p*-value greater than 0.05 (*p* << 0.001, paired t-test). Furthermore, these models recorded nested external testing C-indexes of 0.80 ± 0 and 0.59 ± 0.03, respectively. Meanwhile, there was no significant performance difference between the combinations of PCA and CWGB with either HRF-CT or DRF-CT. The performances of PCA with FSVM or RSF on HRF-CT or DRF-PET were similar and below 0.65. All nested external testing performances are shown in Appendix A.

The fact that the prediction algorithms showed similar but poor performances with DRFs and HRFs in both the SL and SSL strategies while the others performed well suggests issues with algorithm compatibility, model complexity, or parameter tuning. The successful algorithms better utilized features and handled data noise, highlighting the need for the optimization of the underperforming models. Figure 7 illustrates the Kaplan–Meier survival curves for the most effective PCA + CWGB + HRFs and DRFs (extracted from CT images). Additionally, the HRFs or DRFs derived from CT significantly outperformed those extracted from PET (both *p* << 0.001, paired *t*-test). Moreover, additional Kaplan–Meier survival curves from the other HMLSs are shown in Appendix A.

## 4. Discussion

LCa is the leading cause of cancer deaths globally, imposing significant economic burdens [71,72]. Accurate OS prediction, influenced by cancer type, stage, patient health, and treatment response [73,74], is essential for treatment decisions, early diagnosis, and personalized therapy [75,76,77]. However, identifying reliable survival biomarkers remains challenging. Techniques like DL [77], radiomics [75], and gene expression signatures [73,76] use clinical, imaging, and molecular data for prognostic models but face issues with data quality, missing values, and interpretability. Enhanced research is needed to improve OS prediction in LCa. SSL, which combines labeled and unlabeled data, is prevalent in NLP and bioinformatics but limited in medicine due to data quality and privacy concerns [28,78,79,80,81]. A review of 887 studies identified 169 prognostic factors for NSCLC, leading to a 12-factor survival model [82]. Additionally, an ensemble machine learning approach using metabolomic biomarkers from tumor biopsies accurately predicted patient survival, highlighting potential targets for improving outcomes [73].

Multiple ML approaches have been developed to predict OS in LCa [28,83]. Radiomics-based methods [30] extract features from CT images, enhancing survival prediction and biomarker identification. Wang et al. [77] and Zhao et al. [84] showed that Adaboost regression, combined with clinical, genomic, and gene expression data, outperforms other ML techniques in terms of accuracy, precision, sensitivity, and specificity for predicting LCa survival and tumor stage. A study [85] found that Adaboost and Random Forest (RandF) models excel in predicting 5-year survival based on health-related quality of life. Dimension-reduction techniques improve prediction accuracy and prevent overfitting by selecting fewer features, with recent work achieving over 82% accuracy [86]. Stacked-ensemble meta-learners further enhance performance, achieving a 0.9 improvement in classifying short versus long survival times [73]. Advanced models include Yang et al.’s [87] long short-term memory DL using wearable activity data, Gupta et al.’s [88] ensemble of decision tree classifiers with clinical and demographic data, and Tajbakhsh et al.’s [89] semi-supervised multi-task learning on CT scans, which improved segmentation and reduced false positives.

In this study, we demonstrated that the SSL strategy significantly outperformed the SL approach in both HRFs (*p* = 0.01, paired *t*-test) and DRFs (*p* << 0.001, paired *t*-test). For example, HRF-PET combined with PCA and MLP in the SSL method achieved an average accuracy of 0.77 ± 0.10, compared to 0.70 ± 0.07 using PCA and XGB in the SL approach. Similarly, DRF-PET in the SSL strategy with PCA and MLP achieved an average accuracy of 0.85 ± 0.05, outperforming the supervised DRF-CT approach, which achieved 0.65 ± 0.08. These findings show that DRF-PET in the SSL strategy offered the highest predictive accuracy. Additionally, the DRF frameworks consistently outperformed the HRF frameworks, with DRF-PET recording the highest accuracy of 0.85 ± 0.05, significantly better than HRF-PET at 0.77 ± 0.10.

HNCa and LCa share clinical and biological similarities, including tobacco and alcohol use, genetic mutations, similar tumor microenvironments, and treatment responses [18,19]. Approximately 70–80% of HNCa cases are linked to tobacco use, increasing the risk of second primary cancers like LCa [20]. Both cancers frequently involve SCC with comparable radiomic features [21,22]. Leveraging these similarities, HNCa datasets can augment LCa datasets within an SSL strategy. Although traditional SSL approaches with similar unlabeled data reduce the need for large labeled datasets, access to large amounts of these similar data, even unlabeled, might be limited or difficult. Our SSL strategy overcomes this limitation by incorporating diverse data, effectively addressing data size constraints across different classification tasks, as demonstrated in this study. Furthermore, DL networks like autoencoders automatically extract complex and unknown features from raw data, whereas handcrafted radiomic methods rely on predefined features that may miss critical information requiring extensive domain knowledge. Consequently, DRFs surpass HRFs in classification tasks by providing more comprehensive data representations and improved prediction performance. Importantly, in the HDTS tests, DRFs from PET and CT showed significant differences between the high- and low-risk groups (*p*-values of 0.023 and 0.016, respectively), while no significant differences were observed for HRFs. This might indicate that CAs performed better with DRFs than with HRFs. By integrating SSL to minimize large data requirements with DRF extraction for comprehensive imaging analysis, our approach reduces reliance on costly and invasive modalities like PET, enabling CT alone to achieve high predictive performance. Furthermore, incorporating DLFs and HRFs did not enhance prediction performance compared to using DLFs or HRFs alone, as the distinct feature types can introduce redundant or conflicting information that may confuse the model and lead to overfitting.

In the survival prediction tasks, the SSL approach was unsuitable due to the requirement of follow-up times, so we employed an SL approach using 199 LCa patients. Here, the CT-based models using DRF and HRF frameworks showed superior performance compared to the PET-based models (*p* << 0.001). The HRF framework achieved an average C-index of 0.79 ± 0.08, while the DRF framework showed a slightly higher C-index of 0.80 ± 0.1, both with highly significant log rank *p*-values below 0.001. Given our comparison studies between DRFs and HRFs, further investigation into the utilization of DRFs as an alternative to HRFs could potentially provide significant enhancements in performance and robustness [90,91].

As many studies have highlighted, data availability and quality are among the primary challenges preventing the widespread implementation of AI in real clinical practice [92,93]. Innovative approaches, such as the SSL strategy used in this study, offer a promising solution to these concerns. By leveraging diverse datasets and pseudo-labeling techniques, this method enhances prediction reliability and generalizability, even in scenarios with limited or heterogeneous data.

Despite promising results, our study has several limitations. The datasets were restricted to specific cancer types (LCa and HNCar) and imaging modalities (CT and PET), which limits the generalizability of our findings. Validation using more diverse datasets, including related cancers such as bladder, kidney, and cervical cancers, is necessary to confirm the broader applicability of this approach. Incorporating clinical data alongside imaging datasets could further enhance the robustness and clinical relevance of the predictions. Another limitation is the interpretability of the results. Future research should prioritize investigating the underlying radiological features and pathophysiological mechanisms associated with diverse cancer datasets. This exploration will help ensure more explainable and clinically meaningful outcomes. By addressing these limitations, researchers can achieve a deeper understanding of the model’s performance and enhance its potential for broader clinical application. While DRFs capture complex patterns, they are less interpretable than HRFs, and utilizing them results in improved performance but reduced interpretability. On the other hand, the SSL strategy, reliant on pseudo-labeling, may introduce noise, emphasizing the need for high-quality pseudo-labels. The limited sample size may affect the generalizability of the results, requiring further validation on independent datasets. Although feature selection is a future area of interest, we used PCA to reduce the feature size and prevent overfitting. For survival predictions, the SSL strategy was unfeasible due to the need for the last follow-up date, which cannot be predicted.

For future research, a focus on related cancers, employing similar pseudo-labeling strategies with diverse datasets, could further validate and refine this approach. Additionally, determining the specific contribution of each dataset to improving predictive outcomes would provide valuable insights, guiding the development of even more robust and clinically applicable AI-based solutions for survival prediction and beyond. Also, applying SSL to survival analysis could be valuable if the latest follow-up times or OS data from diverse patient datasets become available.

## 5. Conclusions

This study applied semi-supervised ML with pseudo-labels and compared it to a supervised-only strategy for predicting OS. The key findings include the following: (i) both the CT and PET images performed well in the SSL strategies within the DRF framework, achieving accuracies of 0.83 ± 0.06 and 0.85 ± 0.05, respectively, using PCA and MLP; (ii) the DRF frameworks outperformed the HRF frameworks, with DRFs achieving 0.85 ± 0.05 versus HRFs achieving 0.77 ± 0.10; also, combining DRFs and HRFs did not improve OS prediction beyond DRF-PET in the SSL strategy, which achieved a value of 0.85 ± 0.05; (iii) the SSL strategy (PCA, MLP, DRF-PET) significantly outperformed the SL strategy (PCA, KNN, DRF-CT) with an accuracy of 0.85 ± 0.05 versus 0.65 ± 0.08. This study showed that integrating SSL with DRFs reduces dependence on costly modalities like PET, enabling CT alone to achieve high predictive performance. In the survival analysis, the HRF and DRF frameworks using CT images and PCA + CWGB showed superior performance compared to PET, achieving a C-index of ~0.80 with highly significant log rank *p*-values (<0.001). These findings demonstrate the potential of this approach for clinical implementation, particularly in addressing challenges related to the availability of imaging and patient datasets in single or multiple clinical centers. By leveraging SSL with diverse and pseudo-labeled datasets, this approach enhances the reliability and generalizability of survival predictions. The algorithm’s ability to integrate diverse datasets improves robustness, enabling its application in clinical settings where limited data may otherwise hinder predictive performance. This strategy paves the way for more accurate and reputable results, supporting evidence-based decision-making and optimizing lung cancer patient care through better survival predictions using PET/CT imaging.

## Figures and Tables

**Figure 1 cancers-17-00285-f001:**
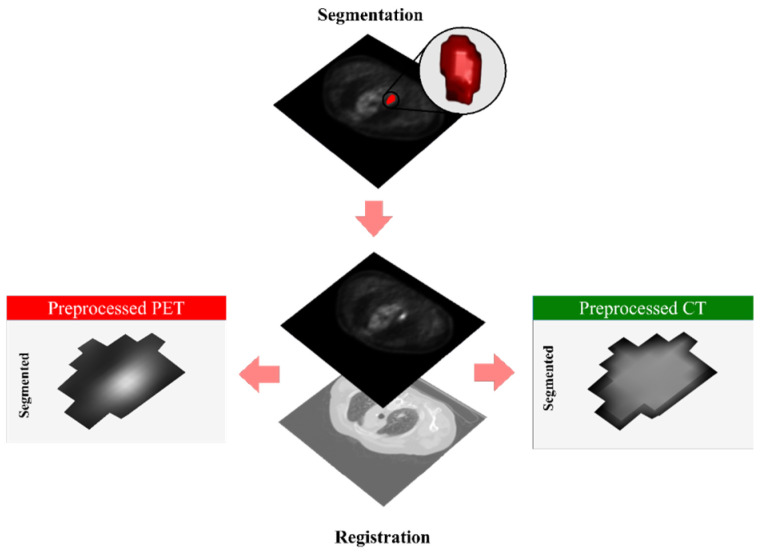
Example of PET, CT, and segmented lung cancer area.

**Figure 2 cancers-17-00285-f002:**
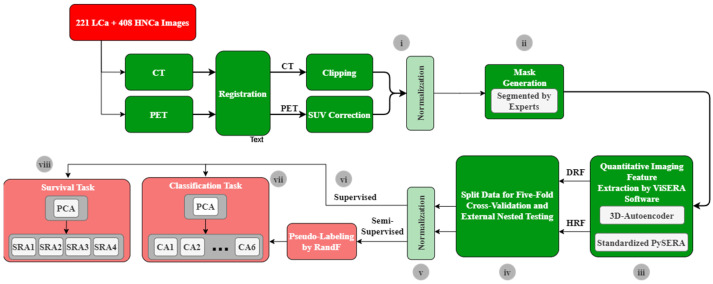
A schematic diagram of the proposed workflow. The study procedure included (**i**) an image preprocessing step, (**ii**) mask generation, (**iii**) imaging feature extraction from the mask applied to preprocessed PET and CT images, (**iv**) splitting the LCa dataset for five-fold cross-validation and external nested testing, (**v**) normalizing HRFs and DRFs, (**vi**) supervised and semi-supervised prediction tasks, (**vii**) PCA linked with 6 CAs, and (**viii**) PCA linked with 4 SRAs. SUV: Standardized Uptake Value, DRF: deep radiomic feature, HRF: handcrafted radiomic feature, PCA: principal component analysis, CA: classification algorithm, SRAs: hazard ratio survival analysis algorithms, RandF: Random Forest Classifier, LCa: lung cancer, HNCa: head and neck cancer.

**Figure 3 cancers-17-00285-f003:**
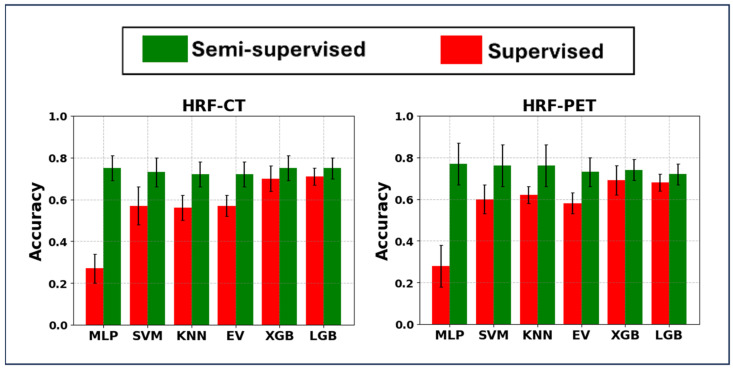
Bar plot of mean ± standard deviation of hybrid machine learning systems (principal component analysis linked with classifiers) when applied on (**left**) HRF-CT, where HRFs (handcrafted radiomic features) extracted the segmented CT images, and (**right**) HRF-PET, where HRFs extracted the segmented PET images. MLP: Multi-Layer Perceptron; SVM: Support Vector Machine, BR: Bagging Regression, KNN: K-Nearest Neighbor, EV: Ensemble Voting Algorithm, XGB: Extreme Gradient Boosting, and LGB: Light Gradient-Boosting Machine.

**Figure 4 cancers-17-00285-f004:**
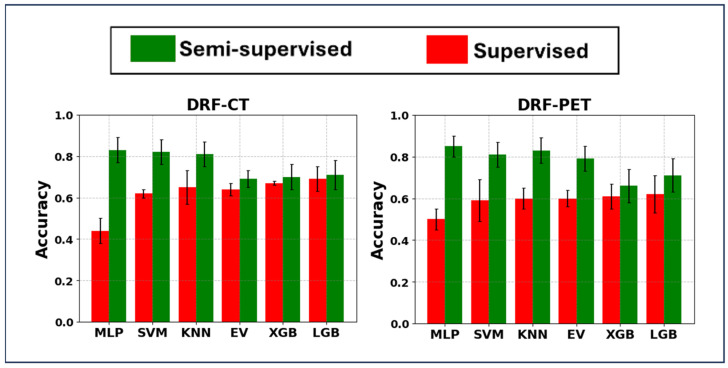
Bar plot of mean ± standard deviation of hybrid machine learning systems (principal component analysis linked with classifiers) when applied on (**left**) DRF-CT, where DRFs (deep radiomic features) extracted the segmented CT images, and (**right**) DRF-PET, where DRFs extracted the segmented PET images. MLP: Multi-Layer Perceptron; SVM: Support Vector Machine, BR: Bagging Regression, KNN: K-Nearest Neighbor, EV: Ensemble Voting Algorithm, XGB: Extreme Gradient Boosting, and LGB: Light Gradient-Boosting Machine.

**Figure 5 cancers-17-00285-f005:**
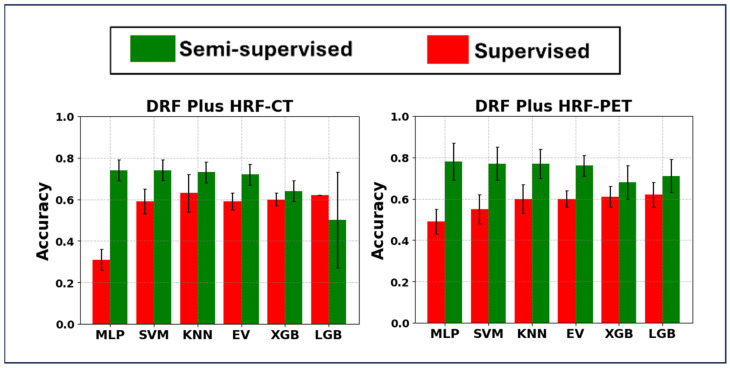
Bar plot of mean ± standard deviation of hybrid machine learning systems (principal component analysis linked with classifiers) when applied on (**left**) mixture of DRF-CT and HRF-CT and (**right**) mixture of DRF-PET and HRF-PET. DRF-CT: DRFs (deep radiomic features) extracted the segmented CT images; HRF-CT: HRFs (handcrafted radiomic features) extracted the segmented CT images; DRF-PET: DRFs extracted the segmented PET images; HRF-PET: HRFs extracted the segmented PET images. MLP: Multi-Layer Perceptron; SVM: Support Vector Machine, BR: Bagging Regression, KNN: K-Nearest Neighbor, EV: Ensemble Voting Algorithm, XGB: Extreme Gradient Boosting, and LGB: Light Gradient-Boosting Machine.

**Figure 6 cancers-17-00285-f006:**
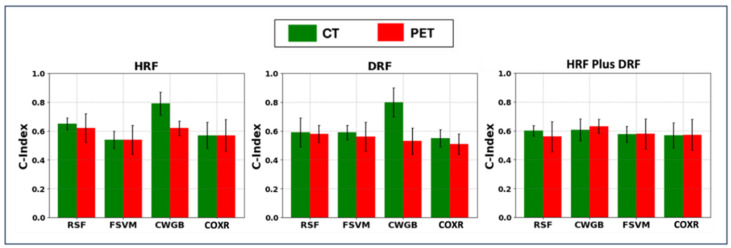
The performances provided by hybrid machine learning systems: principal component analysis (PCA) linked with Fast Survival Support Vector Machine (FSVM), Component-wise Gradient Boosting Survival Analysis (CWGB), Random Survival Forest (RSF), and Cox’s regression (COXR). The four existing datasets included HRFs (handcrafted radiomic features) and DRFs (deep radiomic features) extracted the segmented CT and PET images, respectively.

**Figure 7 cancers-17-00285-f007:**
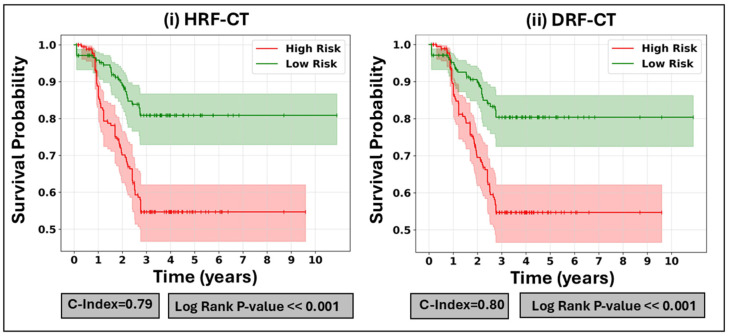
Variation in Kaplan–Meier survival curves generated by the top-performing hybrid machine learning systems (HMLSs) including PCA + CWGB applied on (i) HRF-CT (HRFs derived from CT images) and (ii) DRF-CT (DRFs derived from CT images). PCA: principal component analysis, HRFs: handcrafted radiomic features, DRFs: deep radiomic features, CWGB: Component-wise Gradient Boosting Survival Analysis.

**Table 1 cancers-17-00285-t001:** Demographic and clinicopathologic features of different datasets.

Dataset	Lung Cancer Data from TCIA	Lung Cancer Data from BC Cancer	Head and Neck Cancer Data from TCIA	All
Age	65.24 (13.20)	68.68 (9.24)	62.35 (9.65)	64.24 (10.16)
Sex	19 female - 14 male	81 female - 85 male	88 female - 320 male	188 female - 419 male
N-sample	33 (30 survivor) (3 non-survivor)	166 (127 survivor) (39 non-survivor)	408	607
Histology	Adenocarcinoma (29), squamous cell carcinoma (2), non-small-cell carcinoma (NOS) (2)	Acinar cell carcinoma (3), adenocarcinoma (79), adenosquamous (3), squamous cell carcinoma (46), combined small-cell carcinoma (1), neuroendocrine (3), large-cell carcinoma (1), non-small-cell carcinoma (NOS) (22)	Squamous cell carcinoma (408)	Adenocarcinoma (108), squamous cell carcinoma (456), non-small-cell carcinoma (NOS) (24), acinar cell carcinoma (3), adenosquamous (3), combined small-cell carcinoma (1), neuroendocrine (3), large-cell carcinoma (1),
Stage at diagnosis	Unknown (1), IA (4), IB (4), IIB (1), IIIA (2)	IA (28), IB (52), IIA (54), IIB (32)	I (20), II (34), IIB (4), III (78), IV (57), IVA (188), IVB (23), IVC (4)	Unknown (1), I (20), IA (32), IB (56), II (34), IIA (54), IIB (37), III (78), IIIA (2), IV (57), IVA (188), IVB (23), IVC (4)

## Data Availability

Restrictions apply to the availability of these data. Lung cancer data were obtained from the BC Cancer Agency and are available [from http://www.bccancer.bc.ca/] with the permission of the BC Cancer Agency on 20 April 2023. Other public datasets were provided from The Cancer Imaging Archive (TCIA, https://www.cancerimagingarchive.net/collection/tcga-lusc/). All code (including predictor and dimension reduction algorithms) is publicly shared at: https://github.com/MohammadRSalmanpour/Semi-Supervised-approach-for-servival-prediction-in-Lung-cancer-/tree/main.

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
