# Peer review of "Enhanced Lung Cancer Survival Prediction Using Semi-Supervised Pseudo-Labeling and Learning from Diverse PET/CT Datasets"

_cancers, 2025, doi:10.3390/cancers17020285_

Round 1

Reviewer 1 Report

Comments and Suggestions for Authors

Overall this is an interesting study with a clinical applicability. I have a few comments:

1. Some additional data regarding study participants is required. It is difficult to understand whether generally this procedure is promising for all LCs or subtypes. It will be interesting to give a proof or description including LCs subtypes and staging.

2. p-values should by replaced by only p. Additionally, exact p values should be presented instead of p<0.05 or p<0.01.

4. Please better concluded, how does this algorithm can improve clinical practice.

5. Limitations of te study should be better explained.

Reviewer 2 Report

Comments and Suggestions for Authors

The article presents an interesting approach to building a new machine learning model supporting the analysis of cancer patient survival. Below are some of my comments and suggestions that could improve the quality of this manuscript.

1)       The title of the article talks about the analysis of survival of patients with lung cancer. However, in the abstract you can read about head & neck cancer. This requires explanation.

2)     I would add an additional keyword: survival prediction

3)       I propose moving Table 1 from the supplementary materials to the article because it contains important information. I would also expand this table to clearly show how many data samples are in class 1 - survived at least 2 years and class 2 - died within 2 years.

4)       It would be worth extending the experiment for supervised algorithms with some algorithm from the gradient boosting group. Such methods are characterized by high accuracy. One could use e.g. XgBoost or LightGBM

5)     The authors reduced a large number of dimensions to only 10. What % of the variance is explained by these 10 dimensions in the different experimental scenarios that were conducted?

6)     Authors in their work very often use various abbreviations. It would be good to introduce an appropriate section where these abbreviations are explained collectively.

7)     In my opinion, for the best models achieved in individual experiments, it would be worth preparing error matrices and ROC curves and including them in the article. The classification accuracy result alone does not fully reflect the model's capabilities.

Reviewer 3 Report

Comments and Suggestions for Authors

I have thoroughly reviewed the article “Enhanced Lung Cancer Survival Prediction using Semi-Supervised Pseudo-Labeling and Learning from Diverse PET/CT Datasets” and I don't to find relevance in the authors' conclusion.

They state that “shifting from HRFs handcraft and SL supervised learning to DRFs deep learning radiomics features and SSL semi-supervised learning strategies, particularly in contexts with limited data points, enabling CT or PET alone to significantly achieve high predictive performance” which, in my experience, is pretty obvious. Since their introduction deep learning and semi-supervised learning strategies have always shown higher predictive performance over handcraft and supervised learning (as an example of reference I suggest https://ieeexplore.ieee.org/abstract/document/8675097). It is also misleading that this is true “particularly in contexts with limited data points”. In contexts with limited data points, it is much more likely to have overfitting of the data (especially with DL), thus reducing the generalizability of the results. It is well known that DL requires more data than ML to work properly; it does not require less data time points.

So, the conclusions are partly obvious and partly not properly correct thus hindering the entire meaning of the article and the methodology applied which lacks cross-control and balancing over performance and overfitting/generalizability.
